# Programmable UV-Curable Resin by Dielectric Force

**DOI:** 10.3390/mi14020490

**Published:** 2023-02-20

**Authors:** Yi-Wei Lin, Chang-Yi Chen, Ying-Fang Chang, Yii-Nuoh Chang, Da-Jeng Yao

**Affiliations:** 1Institute of NanoEngineering and Microsystem, National Tsing Hua University, Hsinchu 30013, Taiwan; 2Mechanical and Mechatronics System Research Laboratories, Industrial Technology Research Institute, Hsinchu 310401, Taiwan; 3Department of Power Mechanical Engineering, National Tsing Hua University, Hsinchu 30013, Taiwan

**Keywords:** dielectric force, UV curable resin, programmable UV resin

## Abstract

In this study, UV-curable resin was formed into different patterns through the programmable control of dielectric force. The dielectric force is mainly generated by the dielectric chip formed by the interdigitated electrodes. This study observed that of the control factors affecting the size of the UV resin driving area, current played an important role. We maintained the same voltage-controlled condition, changing the current from 0.1 A to 0.5 A as 0.1 A intervals. The area of droplets was significantly different at each current condition. On the other hand, we maintained the same current condition, and changed the voltage from 100 V to 300 V at 50 V intervals. The area of droplets for each voltage condition was not obviously different. The applied frequency of the AC (Alternating Current) electric field increased from 10 kHz to 50 kHz. After driving the UV resin, the pattern line width of the UV resin could be finely controlled from 224 um to 137 um. In order to form a specific pattern, controlling the current and frequency could achieved a more accurate shape. In this article, UV resin with different patterns was formed through the action of this dielectric force, and after UV curing, tiny structural parts could be successfully demonstrated.

## 1. Introduction

Coplanar interdigitated finger electrodes can be used for a di-electrowetting system [1,2,3,4,5,6]. The design of electrodes was different with conventional solid square or rectangle shapes for EWOD (electrowetting-on-dielectric) system. Although the behavior in the contact angle change in the di-electrowetting system is apparently similar to EWOD system [7,8,9,10], its physics and mechanism are quite different.

MacHale’s team introduced a di-electrowetting concept that is based on liquid di-electrophoresis (L-DEP) [11,12,13]. They used coplanar interdigitated finger electrodes for distinguishing di-electrowetting. Identically, Jones’s also introduced a similar concept through L-DEP for microfluidics application [14,15,16,17,18,19,20,21,22,23]. The driving force of di-electrowetting is the liquid dielectrophoretic force that is generated from charge diploes induced by a non-homogeneous electric field. Two adjacent electrode fingers will be generated by non-homogeneous fringe fields when a voltage is applied.

Brown’ s team [24,25,26] demonstrated a periodic wrinkle that is formed by dielectrophoretic actuation. The thickness of the liquid was very thin and was less than the penetration depth. This means that the fringe electric field would affect the top liquid–vapor interface.

In 2009, Fair et al. proposed a scale model of a dielectric wetting actuator [27]. By considering contact-angle hysteresis, the resistance of the upper and lower plates and the change in force when driving a droplet to move between the electrodes, the minimum voltage value to drive the droplet was calculated, which was related to the electrode width, the thickness of the dielectric layer, and the distance between the upper and lower plates, which were proportional to the ratio. When the electrode size decreased, the distance moved by a droplet generated by the storage tank thus also decreased, which was helpful for the generation of microdroplets.

In 2013, Hiroyuki Fujita’s team proposed a method to fabricate microfluidic wafers rapidly using dielectric-wetted array wafers [28]. This method injected liquid paraffin into the wafer instead of air as the medium during the first step, then injected deionized water into the array wafer. The final step was to turn on a specific electrode and apply a voltage 50~150 Vrms/50 kHz to drive the liquid beads so as to form the desired pattern. After the pattern was formed, the AC (alternating current) signal was maintained on the selected electrodes so as to maintain the shape of the DI (deionized water) water. The wafer temperature was lowered by thermoelectric cooling of the wafer; the temperature was in the range of +45 °C to −40 °C. The entire solidification took 5 s to complete. With this wafer for microfluidic patterning, by programming the change in wafer temperature, the transition from English letter “C” to “A” could be completed in less than 5 min.

In 2015, Papautsky et al. proposed combining continuous microfluidics and digital microfluidics into a microfluidic platform that could be programmed [29]. This study was helpful for the application of microfluidics and biomedical clinical diagnosis. The chip was composed of programmable two-dimensional array electrodes on the bottom plate, with 5 × 8 electrodes in total; the electrode size was 1 mm × 1 mm, the upper cover and the bottom plate were separated by 100 μm, and a syringe pump was used to inject liquid at the inlet. The same amount of liquid was injected from the inlet into six of the nine cell array electrodes in the wafer; the electrodes were turned on with the letter “U”; after two seconds, the letter “C” formed, and then the letter “T” formed. Finally, the letter “U” formed.

The main purpose of this study is to manipulate UV resin for patterning through dielectric force. UV resin is an adhesive material that can be solidified through exposure to UV light, and it is commonly used a glue for paint, coating, and ink. We also discuss the influence of electric field conditions on UV resin manipulation. Finally, through the dielectric force applied on the UV resin, the patterned arrangement of UV resin is further manipulated. After we patterned the UV resin, applied with UV light, we successfully demonstrated the fabrication of tiny structures.

## 2. Theory and Experimental Setup

### 2.1. Theory of Dielectric Force

If droplets are immersed in the other immiscible dielectric liquid with a low dielectric constant, this phenomenon is called di-electrowetting. The elongation of a droplet could be calculated by the following equation [30]: (1)$$h^{2}=h_{0}^{2}-\frac{\left(\varepsilon_{L}-\varepsilon_{F}\right)V_{0}^{2}\Omega }{4\delta_{\gamma LF}}$$
where $h^{2}$ is the height of the droplet and $h_{0}^{2}$ is the initial height of the droplet, while $V_{0}$ = 0 V, *Ω* = *hl*, *l* is the contact length of the droplet in elongation. *δ* is the penetration depth of the electric field (=4 ω/π). $\gamma_{LF}$ is the interfacial tension between the droplet liquid and immersion fluid. $V_{0}$ is the amplitude of applied voltage to the electrodes. $\varepsilon_{L}$, $\varepsilon_{F}$, and $\varepsilon_{0}$ are the permittivities of the droplet liquid, surround fluid and free space, respectively. The thickness of the droplet is proportional to the square of $V_{0}^{2}$ and the ratio of difference in permittivity to the liquid–fluid interfacial tension ($\varepsilon_{L}-\varepsilon_{F})/\gamma_{LF}$. The dielectric model was shown in Figure 1 [30].

### 2.2. Dielectric Chip Manipulate System

The experimental setup of this research (shown in Figure 2) included a dielectric chip holding platform, a UV-curing module (UV mercury lamp (350–450 nm)), a CCD (Charge-coupled Device) alignment system (Confocal Displacement Sensor, CL 3000, KEYENCE), a signal generator (33220A, Keysight), a power amplifier (A304, A. A. Lab system Ltd., Wilmington, DE, USA), a microscope observation module, load/unload platform, and a vacuum module. The designed dielectric chip was placed on the platform and was connected to the relay. The relay could correspond to 160 sets of independent electrode switches, and the patterned control electrode could drive the UV resin.

After the UV resin was driven by the AC electrical field, the UV and vacuum module picked up the glass substrate through the x-y motion platform with ±0.01 mm repeatability (R138, Yu-Zhan motion technology, Taichung City, Taiwan). The vacuum module was moved to just above the dielectric chip, and was in contact with the patterned UV resin. Then, the UV light was turned on for curing to complete the formation of the first layer structure. This procedure is shown in Figure 2b.

### 2.3. UV-Curable Resin

This research used UV-SIL photosensitive polymer material (Agent company in Taiwan was Heart-bond industrial materials LTD., the manufacturer was Adhetron in USA.) mixed with propylene carbonated (CAS number: 108-32-7) as a dielectric force driving liquid.

UV SIL is a is non-corrosive, single-component silicone elastomer material that will cure to a soft rubber after exposure to UV light [31]. The mixing ratio was 80% UV SIL and 20% propylene carbonate. The main purpose for choosing PC (propylene carbonate) was to ensure the driving liquid would have a good permittivity with 66.1 Tm/A (permittivity), which is in excellent agreement Equation 1. UV resin composed of UV-SIL and propylene carbonate is a kind of adhesive material that is cured with UV light (absorption wavelength was 365 nm) and can be used as a glue for paints, coatings, inks, etc. The principle of UV resin curing is that the photo-initiator (or photosensitizer) in the UV curing material generates active free radicals or cations after absorbing ultraviolet light, which initiates monomer polymerization, cross-linking, and branching chemical reactions. The adhesive changes from liquid to solid within a few seconds. The scheme of reaction of UV resin after UV exposure is shown in Figure 3.

### 2.4. Dielectric Chip Design and Manufacture

The metal-electrode structure was formed through photolithography on a silicon wafer [32]. Spin coating was used to deposit a dielectric insulation layer (the insulating layer material can be silicon nitride, SU8 photoresist, etc.) and hydrophobic layer (the hydrophobic layer material can be Teflon, Cytop, etc.). The manufacture flow is shown in Figure 4a.

As the hydrophobic layer plays an important role in resin patterning, when a dielectric chip is made, the hydrophobic characteristics must be analyzed first and the contact angle must be larger than 90°.

We used an 8-inch silicon substrate to make electrodes for the dielectric chip (shown in Figure 4b), with four sets of electrodes in this 8-inch silicon substrate. The electrode size of each set was 1.2 mm × 1.2 mm, 2.0 mm × 2.0 mm, 2.6 mm × 2.6 mm, 4.6 mm × 4.6 mm. In addition, the interdigital electrode applied in the dielectric chip increased the driving force of the resin, making it easier to move and to control the UV resin. Figure 4c shows that the line width and line distance of each interdigital electrode were 10 µm and 10 µm.

## 3. Results and Discussion

### 3.1. Influence of Current and Voltage on UV Resin Drive by Dielectric Force

The voltage change was set from 100 V to 300 V, at intervals of 50 V. In addition, current conditions are set for each voltage condition, ranging from 0.1 A to 0.5 A. The frequency of the AC electric field was fixed at 40 kHz. The sample volume was controlled at 0.2 μL. As Figure 5 shows, the sample was stretched to a square pattern by four electrodes switched at the same time. The measurement result of the square area was considered as the influence of voltage and current. When the current condition was fixed and the voltage change increased from 100 V to 300 V, the shape of the droplet expanded to a specific area. This means that the voltage increased by 300%, while the shape of the droplet increased by 200%, as shown in Figure 6. Observing the trend of the results shows that the sample stretch was affected by the electric current condition.

Under the fixed voltage condition, the current increased from 0.1 A to 0.5 A. As shown in Figure 7, taking the voltage condition of 150 V as an example, the current increased by 500%, and the shape of the droplet increased by 180%. The resulting trends for changing the current under different specific voltage conditions were similar. According to the data observation, the voltage had a positive exponential relationship with the area.

The final paragraphs provide the pattern result for different driving voltages and currents. We discovered the relationship for the condition of a fixed current. The area of the droplet expanded by 28% for every 10 V increase in voltage before the droplet expanded in the saturation area. On the contrary, under the condition of fixed voltage, the area of droplet expanded by 1 mm^2^ for every 0.1 A increase in current. Based on the above results, in addition to the voltage affecting the area of the droplet, the current also played an important role.

### 3.2. Influence of Electric Field Frequency on the Linewidth after UV Resin Drive

Through control of the electric field, the charge distribution on the surface of the colloid could be further varied. The UV resin could achieve the purpose of patterning. In addition to the voltage and current that can affect the driving of the UV resin, we found that the frequency adjustment of the alternating current could affect the appearance of the line width after the UV resin was driven. This section discusses the relationship between the frequency of the electric field and line width after UV resin was driven. The voltage condition of the test was fixed at 200 Vcc and the current was fixed at 0.3 A. The frequency was adjusted from 10 kHz to 50 kHz at intervals of 10 kHz. The test results in Figure 8a show that when the frequency condition was 10 kHz, the UV resin could be formed along the electrode, but the edge of the UV resin was jagged and could not form a straight line. The line width of the UV resin after driving was larger than the line width of the electrode itself, reaching 224 μm. To make the UV resin after driving appear straight and further decrease the line width, we increased the frequency. The results in Figure 8a–e show that the line edge and line width after UV resin driving could become straighter and the line width became smaller as the frequency increased. When the electric field frequency increased from 10 kHz to 20 kHz, the line edge and line width were obviously different. The minimum line width could reach 137 μm, which was close to the electrode linewidth of 100 μm.

As shown in Figure 9, when the frequency increased from 10 kHz to 50 kHz, the line width of the UV resin after being driven by dielectric force was significantly reduced from 224 μm to 137 μm. The line width of the electrode design was 100 μm, and the difference was reduced from 124 μm to 37 μm. From the results, we can see that under the same voltage and current conditions, increasing the frequency could effectively reduce the line width of the UV resin after dielectric force was applied. This could improve the driving precision of the UV resin.

Through the dielectric system, this study successfully demonstrated that the patterning of UV resin could be performed using electrode control. Curing through UV light and repeated patterning and curing steps produced tiny structural parts of different shapes. As shown in Figure 10, a structure with a thickness of about 1~2 mm could be formed by dielectric force and cured by UV light.

The results of this study were compared with conventional 3D printing techniques, as shown in Table 1. The main difference was that the fabrication of structural electronics could be carried out through the same system in this study. There was an opportunity to create conductive layers inside the structure. In addition, the performance of fineness could be optimized by designing the size of the electrodes and the frequency of the applied electric field. The fineness could reach a micron level. In the future, there will be opportunities to apply structural electronic verification in the RD stage, which can accelerate the time course of product development.

## 4. Conclusions

Dielectric force-drivable UV resin, a dielectric chip, and manipulation equipment were developed in this study. We proposed a control model for the current and voltage of the electric field that drives the UV resin effectively. Current played an essential role in the dielectric system. The system provided a pattern instant curing ability with UV treatment.

Through the pattern curing experiment results, the appearance, area, and length of UV resin were affected by the applied current, voltage, and frequency on the dielectric chip. Therefore, the manipulation of dielectric force could be used as a reference for subsequent applications in the fabrication of tiny structures.

We found that current could significantly control the area of the droplet compared with the voltage. A phenomenon was found, where under a voltage fixed to 150 V, the droplet area was expended from 4.7 mm^2^ to 8.4 mm^2^ with an input current of 0.1 A to 0.5 A. While the current was set to 0.3 A, the droplet area expended within 5.6 mm^2^ to 6.5 mm^2^ with an input voltage of 100 V to 300 V. The test results indicate that the droplet was mainly affected by the current strength of the electric field. We further increased the morphological feature fining by a factor. Through the different frequency conditions, a coarse and fining pattern edge result was displayed. The line edge was relatively straight. The minimum line width attained was 137 μm.

An embedded conductive circuit inside the structure was accomplished through the integration of a mature inkjet printing technology in this system. According to the report by the Global Structural Electronics Market [33], the market will be raising CAGR by 23.8% based on USD 1.5 billion in 2021 to 2030. Automotive, aerospace, and other electronics industries are different application fields for structure electronics. A hybrid system is necessary for the development of structure electronics technology. Therefore, this research has great potential to be integrated with maskless printing technology and to be applied to structure electronics. In the future, there will be opportunities to apply structural electronic verification in the RD stage, which can accelerate the time course of product development.

## Figures and Tables

**Figure 1 micromachines-14-00490-f001:**
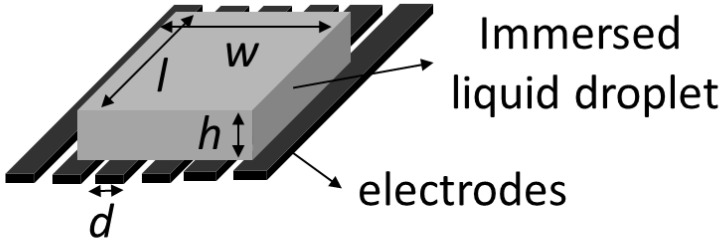
Theoretical analysis of the dielectric model [30].

**Figure 2 micromachines-14-00490-f002:**
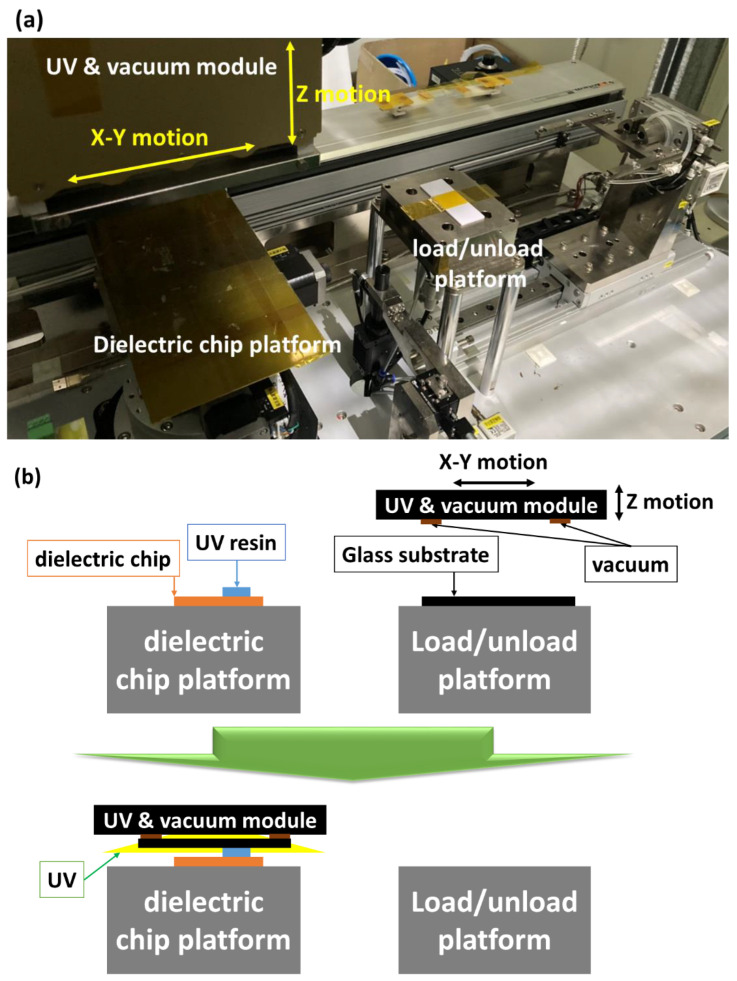
(**a**) Experimental set up of this research. (**b**) Procedure of the dielectric system.

**Figure 3 micromachines-14-00490-f003:**
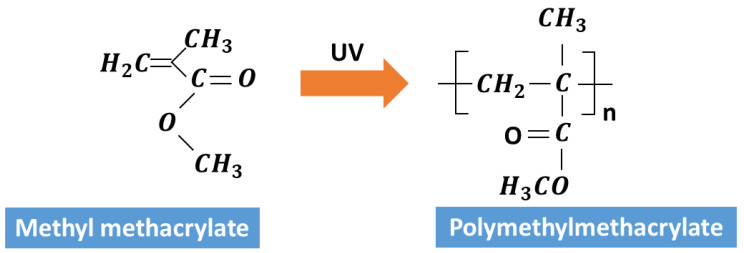
The reaction of UV resin after UV exposure.

**Figure 4 micromachines-14-00490-f004:**
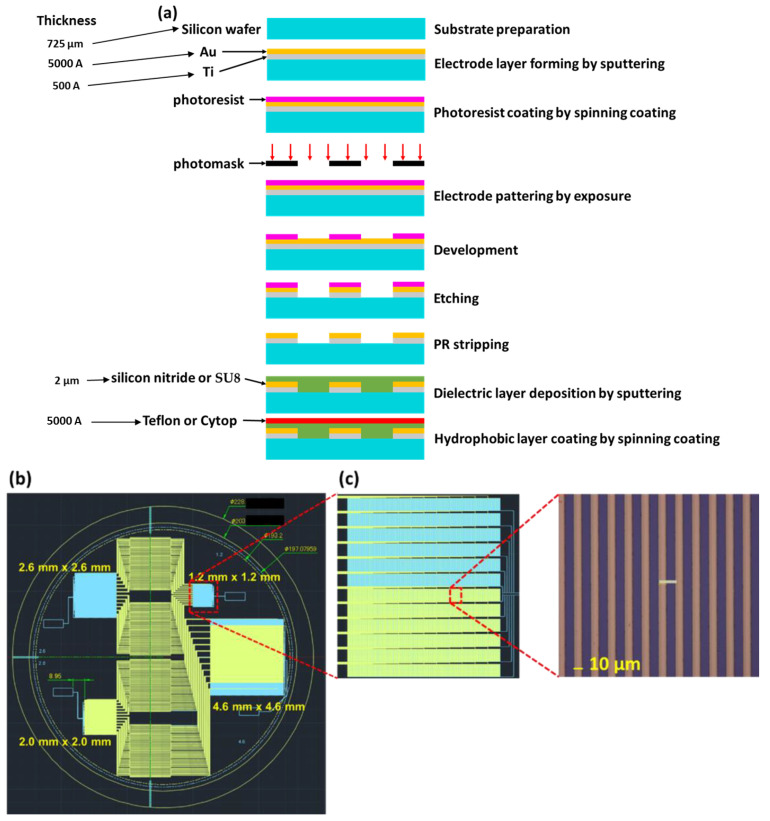
(**a**) Manufacture process flow of dielectric chip. (**b**) Design of the dielectric chip. (**c**) Design of the interdigital electrode.

**Figure 5 micromachines-14-00490-f005:**
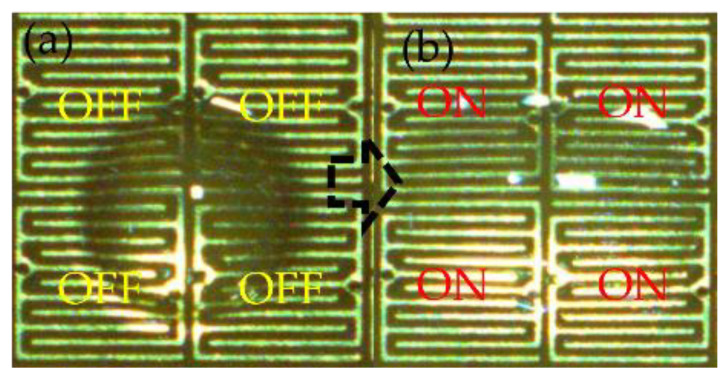
Droplet manipulation control flow. (**a**) Set up the droplet at the center of four electrodes. (**b**) Switch on four electrodes at the same time for the droplet to be stretched.

**Figure 6 micromachines-14-00490-f006:**
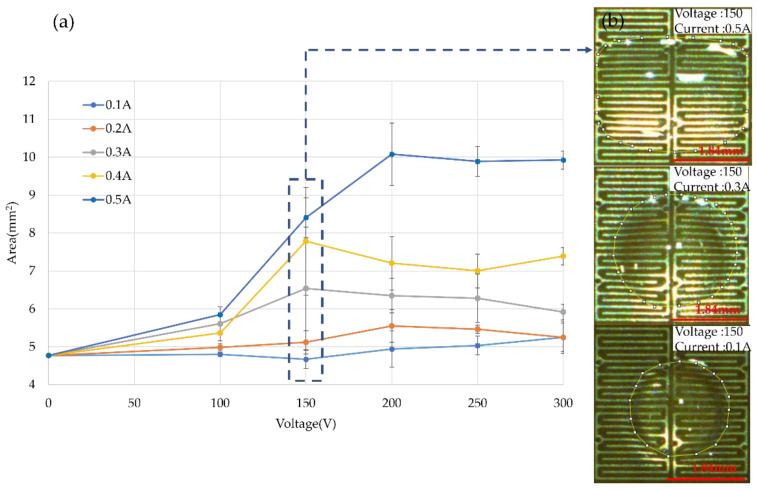
Droplet expanding experiment for current influence (*n* = 5). (**a**) Droplet expanded area result with the current condition fixed. (**b**) Demonstrate the droplet expand situation when the current ranges from 0.1 A to 0.5 A at a voltage of 150 V.

**Figure 7 micromachines-14-00490-f007:**
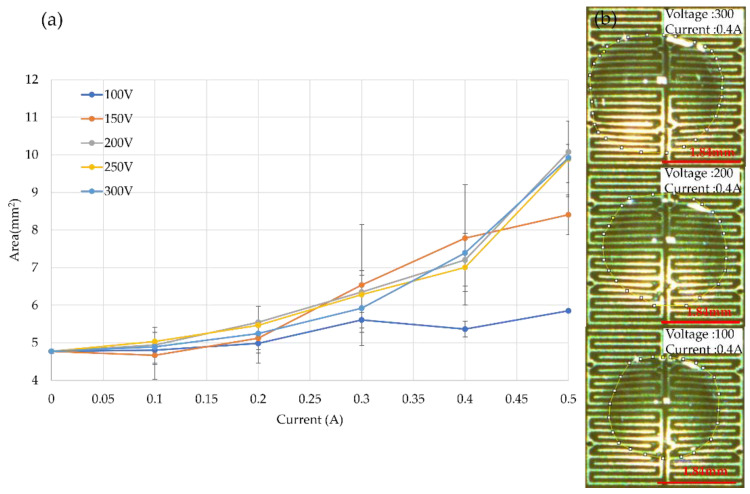
Droplet expanding experiment for voltage influence (*n* = 5). (**a**) Droplet expanded area result with the voltage condition fixed. (**b**) Demonstrate the droplet expand situation when the voltage was from 100 V to 300 V at a current of 0.4 A.

**Figure 8 micromachines-14-00490-f008:**
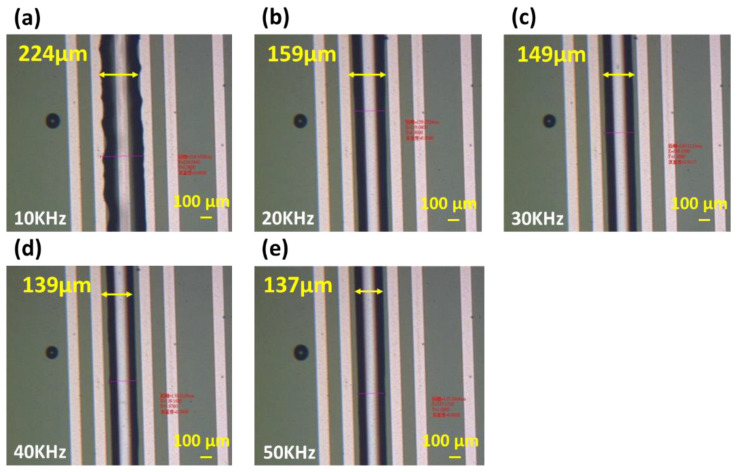
Linewidth of UV resin driven by dielectric force at a frequency of (**a**) 10 kHz, (**b**) 20 kHz, (**c**) 30 kHz, (**d**) 40 kHz, and (**e**) 50 kHz.

**Figure 9 micromachines-14-00490-f009:**
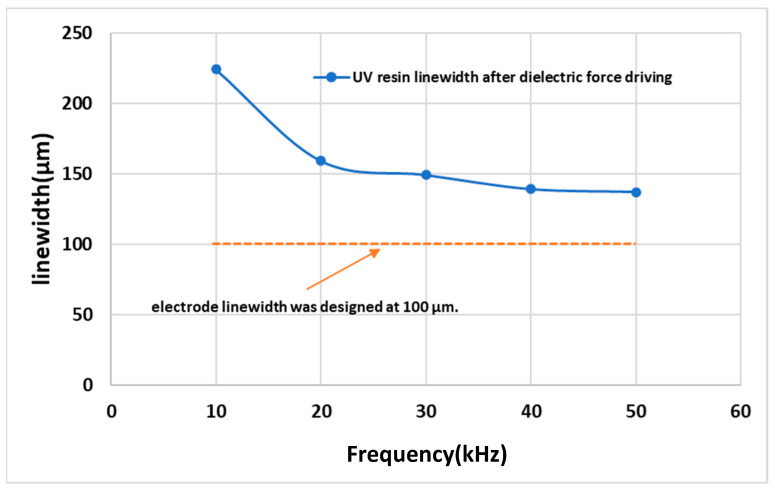
Comparison of electrode linewidth and UV-resin linewidth after being driven. Voltage was 200 V and current was 0.3 A.

**Figure 10 micromachines-14-00490-f010:**
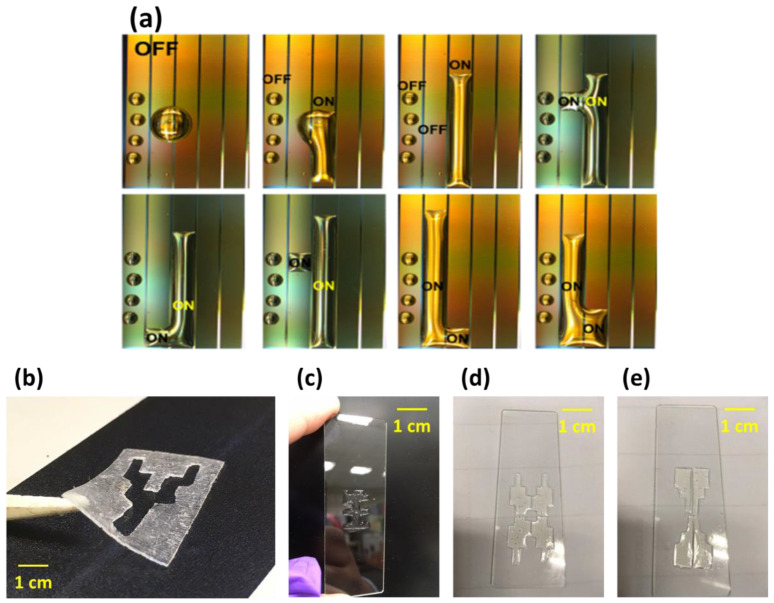
(**a**) UV resin patterning using a dielectric chip. (**b**–**e**) Different shape of structure fabrication using a dielectric system. The thickness of each structure was around 1~2 mm.

**Table 1 micromachines-14-00490-t001:** Comparison of conventional 3D printing and this study.

Technology	This Study	FDM Fused Deposition Modelling	LOM Laminated Object Manufacturing	3DP 3D Printer	DLP Digital Light Processing	SLA Stereo Lithography Apparatus	SLS Selective Laser Sintering	LMD Laser Metal Depostion
Material	Photopolymers, conductive materials	Polymers	Paper and metals	Polymer, metals, and sand	Photopolymers	Photopolymers	Polymers and metals	Metals
Structural electronics fabrication possibility	Yes	No	Maybe	No	No	No	Maybe	no
Accuracy	High	Low	Low	Low	Middle	High	Low	Low

## Data Availability

The data presented in this study are available upon request from the corresponding author.

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
