# Peer review of "Programmable UV-Curable Resin by Dielectric Force"

_micromachines, 2023, doi:10.3390/mi14020490_

Round 1

Reviewer 1 Report (Previous Reviewer 3)

The keyword “additive process” is inappropriate.

Acronyms such as EWOD, AC, DI, CCD, PC have to be explained.

More information devoted to the UV resins should be added to the introductory section of the work.

The Authors should indicate the projected application of the analysed system.

The scheme of curing resin with UV light has to be added.

An explanation is required why only one system consisting of  80% of UV-SIL and 20 % of propylene carbonate was studied.

Figure 3a: the thickness of particular layers has to be presented.

The comparison of the shape obtained using the studied method with other similar procedures has to be presented.

The “Conclusions” section includes observations rather than conclusions and needs to be rephrased.

Author Response

See

Reviewer 2 Report (Previous Reviewer 2)

Review of the manuscript "Programmable UV-curable Resin by Dielectric Force", prepared by a team of 5 people.

Key notes:

1. The manuscript has been prepared for submission to the Special issue "Smart Sensor 2021", the topic of which is quite different from the topic of the article. It is necessary to explicitly explain in the text of the article the connection with this special issue, or send the manuscript to another, more appropriate special issue of Micromachines.

2. In my opinion, in your manuscript it is necessary to clearly indicate the novelty of your work relative to your previous research, clearly showing what new scientific knowledge you demonstrate as a result. The demonstration of several dependences of the parameters of a photocurable polymer material with varying electrical parameters, in my opinion, is insufficient for a “Regular Article” level publication in Micromachines and is more suitable for a “Letters” type publication.

3. “This research using UV-SIL photosensitive polymer material (UV329, Heart-bond industrial materials LTD.) mixed with propylene carbonated (CAS number: 108-32-7) as a dielectric force driving liquid. The mixing ratio was 80% with UV-SIL and 20% with propylene carbonate." - you have an incorrect description of the material you are using. Write a correct description of your material, its manufacturer. Filling silicone rubber "UV-SIL" (I understand correctly that you are using UV-SIL RTV 3011 produced by Adhetron?) contains not only the photoinitiator "(UV329, Heart-bond industrial materials LTD.)", as you can conclude from your text. In addition, you should add a rationale in the text for why you are using this particular material as the main component of the mixture, and not another. For example, you can give its properties in the text (and not the properties of the finished composition), indicating its parameters that are important from the point of view of your method (dielectric coefficients and others), describing why you need these numbers, and why you cannot use, for example, ink from 3D printers working on the principle of photocuring. This information will allow you to justify choosing the 80-20% mixing ratio you use to prepare your ink.

4. Section “2.3. Dielectric Chip Design and Manufacture" did you make this algorithm for the first time for this work? If this is not the case, then in my opinion it would be appropriate to place a link to works where you use this approach.

5. "The sample volume is controlled at 0.2??." - please clarify by what method and with what accuracy you applied and controlled such a volume of liquid sample. I think this is useful information. In my opinion, applying such volumes with the help of automatic pipettes is quite difficult - there can be too much variation in volume with different drops. How critical is this in your work?

6. Graph in fig. 8 is not correct in my opinion. On it, you display the "electrode linewidth" value as a measured value (by measurement points), while it is a constant and is not a variable parameter during the experiment. You need to re-arrange the graph so that it can be interpreted correctly.

7. You need to carefully read the text before sending. The text of the publication contains errors, for example 140 « The voltage change is set from 100 V to 30070 V, at interval 50 V. In addition, current... »

In the current version, the article needs additional significant revision with re-review.

Author Response

See attachment. 

Reviewer 3 Report (New Reviewer)

Review report for “Programable UV-curable Resin by Dielectric Force” by Lin et al. submitted to Micromachines.

This article reports on a systems study of the manipulation of UV-curable resin  by dielectric force. Photolithography was applied to create dielectric EWOD chips for droplet manipulation control flow. Droplet expansion was demonstrated by increasing the voltage or current and associated frequency.  The line width was controlled several-fold. The study covers microfabrication, experimental demonstration, systems parameter analysis, and discussion. Overall, I support the acceptance of the manuscript if the following aspects can be addressed.

1)     The demonstration of creating arbitrary patterns is impressive.  Will it be possible to create numbers like six and eight? Some discussion on the limitations and potential solutions will be useful.

2)     What is the cost of the fabrication? Due to recent inflation globally, some services have increased their price 10-fold (e.g., from $20 to $200). It will be interesting to see this process provide a cost-effective approach to pattern formation.

3)     Is there a theoretical model that predicts the voltage and frequency dependence? Some discussion on this aspect will be awakening in the field and help future investigations of the topic.  

Round 2

Reviewer 1 Report (Previous Reviewer 3)

I have to admit that the Authors have significantly improved the submitted manuscript.

The conclusions, however, still have to be rephrased. More emphasis is required in relation to the actual outcome of the research, deduced from the obtained results, rather than a comparison with other publications. Consequently, Table 1 has to be moved to the section devoted to the discussion of the results.

Author Response

See attachment. 

Reviewer 2 Report (Previous Reviewer 2)

Review of the corrected version of manuscript "Programmable UV-curable Resin by Dielectric Force", prepared by a team of 5 people.

 The authors have corrected the manuscript, but a few corrections raise big questions for me.

 1. Answer to the first question of the reviewer “This study is preliminary research for microstructure fabrication. It is also has highly potentialto integrated with maskless printing technology and apply to structure electronics. According to the report of Global Structural Electronics Market [32], the market will be raising by CAGR of 23.8% based on USD 1.5 billion in 2021 to 2030. Automotive, aerospace, and other electronics industries were application fields by structure electronics. Hybrid system is necessary for the development of structure electronics technology. Therefore, the research in this study is quite important. Embedded conductive circuit inside the structure will be accomplished by integration of mature inkjet printing technology and the system of this study.” - Add at least a 1-2 of suggestions here on how this can be useful in the manufacture of microsensor devices, preferably with a link. You need an _explicit_ justification for linking your research topic to the Special Issue "Smart Sensor 2021" theme.

 2. “This research using UV-SIL photosensitive polymer material (UV329, Heart-bond industrial materials LTD.) mixed with propylene carbonated (CAS number: 108-32-7) as a 113 dielectric force driving liquid. UV SIL is consisting of photoinitiator, silicone and acrylic."

Your description of UV-curable UV-SIL is incorrect. UV329 is a photoinitiator manufactured by Heart-bond industrial materials LTD. Write it as a separate sentence.

 "UV SIL is consisting of photoinitiator, silicone and acrylic." - Such a description is completely unsuitable for a scientific publication. You need to confirm this data with a link to information where the composition is given (for example, to the manufacturer's web page with a description of this product). If you are using a commercially available material with an unknown composition, then write so, indicating the correct name of the manufacturer.

 3. Why do you bring a trivial figure 3? What does the scheme of photopolymerization of methacrylate under ultraviolet light have to do with the topic of the article? You are working with a complex composition with components unknown to you. What actually happens in your material is still unknown. Figure 3 not need here.

Round 3

Reviewer 2 Report (Previous Reviewer 2)

The authors have made adjustments to the manuscript of the article, which allow me, from a formal point of view, to admit the possibility of publication in the journal.
However, I strongly recommend to carry out additional work on the text of the article under the supervision of the Editors of the journal Micromachines, in order to correct technical errors and bring the manuscript into proper condition.

This manuscript is a resubmission of an earlier submission. The following is a list of the peer review reports and author responses from that submission.

Round 1

Reviewer 1 Report

The current manuscript by Lin et al. demonstrates how UV-curable resin can be formed into different patterns through programmable control of dielectric forces. While the general subject could be interesting to the readership of Micromachines, this paper cannot be published in its current form as the novelty isn’t clear. Additionally, it requires very substantial additional work and editing.

My main concerns are:

·         Novelty and importance – I do not understand what is new here. The authors state that “The main purpose of this study is to manipulate UV resin for patterning through dielectric force.”. This has been done before by the same group (not cited) in the paper “Virtual Stencil for Patterning and Modeling in a Quantitative Volume Using EWOD and DEP Devices for Microfluidics”  https://www.mdpi.com/2072-666X/12/9/1104 . What is new here?   There are also many papers on dielectric patterning of liquids on surfaces (with or without microfluidics). Using UV curable resin doesn’t give any new insights that the reader can gain from the current manuscript. Another very similar paper from the same group (not cited) is https://doi.org/10.1063/5.0012684 . The current paper seems like repetition of previous works and doesn’t explain what are the challenges that it overcomes.  

·         Low scientific level – this paper should not have been submitted in the current form. As an example, one major flaw are the figures and especially graphs (Figures 5, 7 and 10). There are no error bars! Was each experiment conducted only once? Additionally, Figure 10 has only two data points for each set and has no labeling of the X axis.

·         Language – I recommend that a professional English editor revise the manuscript. It is very hard to understand what the authors are claiming due to the poor language level.

Reviewer 2 Report

A manuscript of an article entitled "Programmable UV-curable Resin by Dielectric Force" is presented for review in the Micromachines magazine, in the Special Issue "Smart Sensor 2021".

First of all, I can note that the topic of the special issue of the journal does not coincide with the topic of the submitted article. Please clarify your choice of special edition. I do not observe a direct connection with sensory systems.

The work demonstrates a rather original approach to the formation of structures from photocurable materials used in 3D printing. However, the article in the current version has a number of shortcomings that need to be improved.

Methodical part:

1. "After driving the UV resin, the pattern 18 line width of the UV resin can be finely controlled from 224 um to 137 um" - what is the accuracy of the width measurements? exactly 1 micron? When I look at figure 6, I see single measurements with the help of digital, but I do not observe the measurement statistics. This is especially true of Figure 6a where the uneven contour of the side wall of the line. It is unacceptable from the point of view of correct methodology to show data in this form. It is necessary at least to use measurement statistics and errors in the analysis of optical photographs.

2. Much of your modifications are based on measuring quantities (distance and time) from data using optical photography and video. In my opinion, these values ​​will be highly dependent on the amount of polymer applied to your chip. Also, I don't see data on how much polymer to apply, thickness (which should also be measured statistically), or a clear reason why you don't.

3. I don't understand the description of the photocurable resin material used. You provide information about UV-SIL photocurable material and provide UV329 photoinitiator, Heart-bond industrial materials LTD (https://www.echemi.com/sds/2-2-hydroxy-5-tert-octylphenylbenzot-pd1801291012.html - it's him?) . But what exactly is UV-SIL? Who is its manufacturer? What is the absorption spectrum of the finished material? Or at least what are the manufacturer's recommended cure times? In my opinion, Figure 3, in which you illustrate the polymerization process (by the way, is not entirely correct) is generally superfluous here. It is better to characterize the photocurable material in detail.

4. I don't see the scale in all the photos.

5. Presentation of data in figures is not optimal. There is a lot of free space with white spots, and the drawings themselves are small and the measurement results are hard to see on them.

Results:

6. Why is figure 5 needed? The data is in the tables below.

7. Why is table 2 needed? Arrows with the same text can be placed in Table 1, or described in the text.

8. Why is Figure 10 needed as a separate graph? Demonstration of 4 measured points in this version is superfluous. If you really want to present it, then you can combine it with a drawing.

9. “Different shape of structure fabrication by dielectric system. The thickness of each structure was around (1~2) mm" - did you measure the thickness of your polymer in the 1-2mm range? In my opinion, this is unacceptable, including from the point of view of 3D printing. I recommend consulting with specialists who are engaged in 3D printing on photocurable printers, or with people who can print 3D structures on ordinary, cheap 3D printers. Explore the choice of layer thickness in 3D printing. It is of great importance.

These are only the main remarks that need to be eliminated so that the manuscript can be considered and analyzed. Unfortunately, in the current version, it is more like a student report, due to obvious methodological errors as well.

Please improve the manuscript. Based on the results you get, you can make a good publication, but you need to correctly and accurately present what you received.

Reviewer 3 Report

Abstract: the novelty of the publication should be highlighted

Introduction: More information devoted to UV resins as well as possibilities of manufacturing UV resins-based materials and their applications has to be presented.

Figure 1. It has to be explained why the droplet does not adopt a spherical shape.

Figure 3. The scheme has to be improved, the amount of substrate and product is unclear.

Section 3.1 Please explain why changes in current and voltage influence the UV resin drive

lines 230-245. The conclusions presented are rather an observation. The text must be rephrased.

Round 2

Reviewer 1 Report

It seems that the authors do not comprehend that it is impossible for a reader to understand the novelty or importance of their manuscript. They are showing a few (lacking) experiments and do not compare to previous literature or explain what is new. In the revised version they just added a paragraph only in the conclusions section where they say something very vague on this matter.

Major revisions are required.

·         In my original review I commented on studying only two data point (0.3A and 0.4A). The authors ignored my comment and just deleted figure 10 where this was notable. Additional experiments are required where at least 5 different currents are studied.

·         The abstract must be re-written to explain what has been previously done and what is new here.

·         The introduction must be re-written and cite previous works of the author (as I requested in my original review) and others as well as explain what was missing (must be specific – no study of current effect, previous volumes used and so on).

·         It is not enough to state that the volume was increased, or currents changed. There must be a discussion regarding the challenges and/or the influence of these changes. This must be mentioned in the introduction, discussion and conclusions.

·         Figures 1 & 2 must be improved.

 I must insist that a professional English editor is used to go over the entire manuscript (authors have ignored my request). One cannot expect that the readers guess what the authors meant to write. See just one example (unfortunately, there are many examples) in the following sentences:

“Furthermore, although the results of this study show that the thickness of the structure is only in the order of mm. However, in the future, through stacking modules with precise alignment function. The real 3D printing needs can be achieved through several stacking processes.”

Reviewer 2 Report

The corrected manuscript of the article entitled "Programmable UV-curable Resin by Dielectric Force", submitted for consideration to the “Micromachines”, in the Special Issue "Smart Sensor 2021" is submitted for re-review.

Unfortunately, in my opinion, the corrected version of the text of the article does not meet the criteria of the Micromachines journal, and the responses to the reviewer's comments are incorrect.

1) I don't see an answer to the question from the previous review regarding the selection of Special Issue "Smart Sensor 2021". You need to justify (including in the text of the article) the connection between the topic of your article and the topic of the Special Issue.

2) The answer to the first question of the last review:

Thanks for the suggestion. The accuracy of linewidth measurement is nm. But we think the error from hand made measurement was around 1 µm. Therefore, in this paper just count the linewidth with digits."

I am forced to conclude that the authors of the article do not possess the basis of the correct measurement technique. This is the basic knowledge that students should learn at the very beginning of their studies. The numbers with an accuracy of 1nm, which the photo analysis program shows you in no way reflect the real values! The program measures the distance between the pixels of a photo (at best), or between places on the computer screen where you clicked on a button on a computer mouse.

You need to take many measurements of the same quantity in different places, get a lot of values, calculate the average value with a confidence interval of errors. This must be done always during scientific work, otherwise you provide incorrect data.

I recommend reading the books «Basics of measuring physical quantities", for example https://ncert.nic.in/textbook/pdf/keph102.pdf

The current option for providing data is incorrect! Phrases "But, we think the error from hand made measurement was around 1 μm" show the incorrectness of the basis of the measurement technique, which, unfortunately, removes all the other advantages of the article.

3) Description of the photocurable material in the article and answer to question 3:

“This research using UV-SIL photosensitive polymer material (UV329, Heart-bond in-108 dustrial materials LTD.) mixed with propylene carbonated (CAS number: 108-32-7) as a 109 dielectric force driving liquid. The mixing ratio was 80% with UV-SIL and 20% with 110 propylene carbonate. The main purpose of choosing PC is to make the driving liquid have 111 good permittivitiespermittivity. 66.1 Tm/A (permittivitiespermittivity) with propylene 112 carbonate is excellent agreement level to Equation 1. The UV resin composing with UV-113 SIL and propylene carbonate is a kind of adhesive material that is cured with UV light 114 (absorption wavelength was 365 nm ) and can be used as a glue for paints, coatings, inks 115 etc. The principle of UV resin curing is that the photo-initiator (or photosensitizer) in the 116 UV curing material generates active free radicals or cations after absorbing ultraviolet 117 light, initiates monomer polymerization, cross-linking and branching chemical reactions. 118 The adhesive is made to change from liquid to solid within a few seconds."

Thanks for the suggestion. This research using UV-SIL photosensitive polymer material (UV329, Heart-bond industrial materials LTD.) mixed with propylene carbonate (CAS number: 108-32-7) as a dielectric force driving liquid. The mixing ratio was 80% with UV-SIL and 20% with propylene carbonate. The intensity of UV lamp is 21,700 mW/cm2. The curing test and condition of UV resin was shown below.”

You still incorrectly describe the photosensitive polymer material you are using in the article. The description "UV-SIL photosensitive polymer material (UV329, Heart-bond industrial materials LTD)" is incorrect and only appears in your previous article.

Obviously you are using "...photosensitive polymer material UV-SIL (Heart-bond industrial materials LTD), which contain UV329 (CAS number: 3147-75-9) as photoinitiator,...".

The current version of the description of the materials is unacceptable! The phrase "The principle of UV resin curing is that the photo-initiator (or photosensitizer) in the UV curing material generates active free radicals or cations after absorbing ultraviolet light, initiates monomer polymerization, cross-linking and branching chemical reactions." is unsuccessful, it is better to say exactly about the features of the curing process of your material (if you know it) and provide links to reviews in which this is described in more detail.

4) How did you draw the curve through the points in Figure 4? Why is it not linear? How do you justify this?

5) Figure 6 - what information does the lower curve (electrode linewidth) carry - should this value change?

6) I also consider the answer to my question about the thickness of the films to be incomplete.

Unfortunately, this is not the whole list of my comments on this article. I can't paint everything as it will take me too long.

Unfortunately, in its current form, the article is not suitable for publication.

The authors need to correctly measure using correct methods and present the results in a good way, supplementing it with a discussion about the obtained dependencies with their detailed description. It is necessary to improve the text of the article, make the correct presentation of all data in the methodological part and the description of the materials used in accordance with the requirements of the journal.

Unfortunately, now the article does not meet the criteria and level of the «Micromachines» journal.